# Generalization Bounds with Minimal Dependency on Hypothesis Class via Distributionally Robust Optimization

**Yibo Zeng**
Columbia University
yibo.zeng@columbia.edu

**Henry Lam**
Columbia University
henry.lam@columbia.edu

## Abstract

Established approaches to obtain generalization bounds in data-driven optimization and machine learning mostly build on solutions from empirical risk minimization (ERM), which depend crucially on the functional complexity of the hypothesis class. In this paper, we present an alternate route to obtain these bounds on the solution from distributionally robust optimization (DRO), a recent data-driven optimization framework based on worst-case analysis and the notion of ambiguity set to capture statistical uncertainty. In contrast to the hypothesis class complexity in ERM, our DRO bounds depend on the ambiguity set geometry and its compatibility with the true loss function. Notably, when using statistical distances such as maximum mean discrepancy, Wasserstein distance, or $\phi$-divergence in the DRO, our analysis implies generalization bounds whose dependence on the hypothesis class appears the minimal possible: The bound depends solely on the true loss function, independent of any other candidates in the hypothesis class. To our best knowledge, it is the first generalization bound of this type in the literature, and we hope our findings can open the door for a better understanding of DRO, especially its benefits on loss minimization and other machine learning applications.

## 1 Introduction

We study generalization error in the following form. Let $\ell : \mathcal{F} \times \mathcal{Z} \to \mathbb{R}$ be a loss function over the function class $\mathcal{F}$ and sample space $\mathcal{Z} \subset \mathbb{R}^d$. Let $\mathbb{E}_P[\ell(f, z)]$ be the expected loss under the true distribution $P$ for the random object $z$. This $\mathbb{E}_P[\ell(f, z)]$ can be an objective function ranging from various operations research applications to the risk of a machine learning model. Given iid samples $\{z_i\}_{i=1}^n \sim P$, we solve a data-driven optimization problem or fit model to give solution $f_{\text{data}}$. The *excess risk*, or optimality gap of $f_{\text{data}}$ with respect to the oracle best solution $f^* \in \arg\min_{f \in \mathcal{F}} \mathbb{E}_P[\ell(f, z)]$, is given by

$$\mathbb{E}_P[\ell(f_{\text{data}}, z)] - \mathbb{E}_P[\ell(f^*, z)] \tag{1}$$

This gap measures the relative performance of a data-driven solution in future test data, giving rise to a direct measurement on the generalization error. Established approaches to obtain high probability bounds for (1) have predominantly focused on empirical risk minimization (ERM)

$$\hat{f}^* \in \arg\min_{f \in \mathcal{F}} \mathbb{E}_{\hat{P}}[\ell(f, z)]$$

for empirical distribution $\hat{P}$, or its regularized versions. A core determination of the quality of these bounds is the functional complexity of the model, or hypothesis class, which dictates the richness of the function class $\mathcal{F}$ or $\mathcal{L} := \{\ell(f, \cdot) \mid f \in \mathcal{F}\}$ and leads to well-known measures such as the Vapnik-Chervonenkis (VC) dimension [1]. On a high level, these complexity measures arise from the

36th Conference on Neural Information Processing Systems (NeurIPS 2022).

need to uniformly control the empirical error, which in turn arises from the a priori uncertainty on the decision variable $f$ in the optimization.

In this paper, we present an alternate route to obtain concentration bounds for (1) on solutions obtained from distributionally robust optimization (DRO). The latter started as a decision-making framework for optimization under stochastic uncertainty [2–4], and has recently surged in popularity in machine learning, thanks to its abundant connections to regularization and variability penalty [5–12] and risk-averse interpretations [13, 14]. Instead of replacing the unknown true expectation $\mathbb{E}_P[\cdot]$ by an empirical expectation $\mathbb{E}_{\hat{P}}[\cdot]$, DRO hinges on the creation of an *uncertainty set* or *ambiguity set* $\mathcal{K}$. This set lies in the space of probability distribution $\mathcal{P}$ on $z$ and is calibrated from data. It obtains a solution $f_{\text{dro}}^*$ by minimizing the worst-case expected loss among $\mathcal{K}$

$$f_{\text{dro}}^* \in \arg \min_{f \in \mathcal{F}} \max_{Q \in \mathcal{K}} \mathbb{E}_Q[\ell(f, z)]. \tag{2}$$

The risk-averse nature of DRO is evident from the presence of an adversary that controls $Q$ in (2). Also, if $\mathcal{K}$ is suitably chosen so that it contains $P$ with confidence (in some suitable sense) and shrinks to singleton as data size grows, then one would expect $f_{\text{dro}}^*$ to eventually approach the true solution, which also justifies the approach as a consistent training method. The latter can often be achieved by choosing $\mathcal{K}$ as a neighborhood ball surrounding a baseline distribution that estimates the ground truth (notably the empirical distribution), and the ball size is measured via a statistical distance.

Our main goal is to present a line of analysis to bound (1) for DRO solutions, i.e.,

$$\mathbb{E}_P[\ell(f_{\text{dro}}^*, z)] - \mathbb{E}_P[\ell(f^*, z)] \tag{3}$$

which, instead of using functional complexity measures as in ERM, relies on the ambiguity set geometry and its compatibility with *only* the true loss function. More precisely, this bound depends on two ingredients: (i) the probability that the true distribution lies in $\mathcal{K}$, i.e., $\mathbb{P}[P \in \mathcal{K}]$ and, (ii) given that this occurs, the difference between the robust and true objective functions evaluated at the true solution $f^*$, i.e., $\max_{Q \in \mathcal{K}} \mathbb{E}_Q[\ell(f^*, z)] - \mathbb{E}_P[\ell(f^*, z)]$. These ingredients allow us to attain generalization bounds that depend on the hypothesis class in a distinct, less sensitive fashion from ERM. Specifically, when using the maximum mean discrepancy (MMD), Wasserstein distance, or $\phi$-divergence as a statistical distance in DRO, our analysis implies generalization bounds whose dependence on the hypothesis class $\mathcal{L}$ appears the minimal possible: The bound depends solely on the true loss function $\ell(f^*, \cdot)$, independent of any other $\ell(f, \cdot) \in \mathcal{L}$.

Note that although there exist generalization bounds that do not utilize conventional complexity measures like the VC dimension [1], they are all still dependent on other candidates in the hypothesis class and therefore, are different from our results. Examples include generalization bounds based on hypothesis stability [15], algorithmic robustness [16], the RKHS norm [17], and generalization disagreement [18]. These approaches all rely on the data-driven algorithmic output $f_{\text{data}}$, which varies randomly among the hypothesis class due to its dependence on the random training data. Therefore, to translate into generalization bounds, they subsequently require taking expectation over $f_{\text{data}}$ ([15, Definition 3 and Theorem 11]; [18, Theorem 4.2]) or taking a uniform bound over $\mathcal{F}$ ([16, Definition 2 and Theorem 1]; [17, Theorem 4.2]). In contrast, our bound depends on the hypothesis class through the deterministic $\ell(f^*, \cdot)$, in a way that links to the choice of ambiguity set distance. To our best knowledge, generalization bounds of our type have never appeared in the literature (not only for DRO but also other ML approaches). We hope such a unique property will be useful in developing better machine learning algorithms in the future, especially in harnessing DRO on loss minimization and broader statistical problems.

Finally, our another goal is to conduct a comprehensive review on how DRO intersects with machine learning, which serves to position our new bounds over an array of motivations and interpretations of DRO. This contribution is in Section 2 before we present our main results (Section 3 - 5) and numerical experiments (Section 6).

## 2 Related Work and Comparisons

DRO can be viewed as a generalization of (deterministic) robust optimization (RO) [19, 20]. The latter advocates the handling of unknown or uncertain parameters in optimization problems via a worst-case perspective, which often leads to minimax formulations. [21, 22] show the equivalence of

ERM regularization with RO in some statistical models, and [16] further concretizes the relation of generalization with robustness.

DRO, which first appeared in [23] in the context of inventory management, applies the worst-case idea to stochastic problems where the underlying distribution is uncertain. Like in RO, it advocates a minimax approach to decision-making, but with the inner maximization resulting in the worst-case distribution over an ambiguity set $\mathcal{K}$ of plausible distributions. This idea has appeared across various disciplines like stochastic control [24] and economics [25, 26]. Data-driven DRO constructs and calibrates $\mathcal{K}$ based on data when available. The construction can be roughly categorized into two approaches: (i) neighborhood ball using statistical distance, which include most commonly $\phi$-divergence [27–31] and Wasserstein distance [32–34, 6]; (ii) partial distributional information including moment [35, 2–4, 36], distributional shape [37–40] and marginal [41–43] constraints. The former approach has the advantage that the ambiguity set or the attained robust objective value consistently approaches the truth [27, 44]. The second approach, on the other hand, provides flexibility to decision-maker when limited data is available which proves useful on a range of operational or risk-related settings [45–47].

The first approach above, namely statistical-distance-based DRO, has gained momentum especially in statistics and machine learning in recent years. We categorize its connection with statistical performance into three lines, and position our results in this paper within each of them.

## 2.1 Absolute Bounds on Expected Loss

The classical approach to obtain guarantees for data-driven DRO is to interpret the ambiguity set $\mathcal{K}$ as a nonparametric confidence region, namely that $\mathbb{P}[P \in \mathcal{K}] \geq 1 - \delta$ for small $\delta \in (0, 1)$. In this case, the confidence guarantee on the set can be translated into a confidence bound on the true expected loss function evaluated at the DRO solution $f_{\text{dro}}^*$ in the form

$$\mathbb{P}[\mathbb{E}_P[\ell(f_{\text{dro}}^*, z)] \leq \max_{Q \in \mathcal{K}} \mathbb{E}_Q[\ell(f_{\text{dro}}^*, z)]] \geq 1 - \delta \tag{4}$$

via a direct use of the worst-case definition of $\max_{Q \in \mathcal{K}} \mathbb{E}_Q[\cdot]$. This implication is very general, with $\mathcal{K}$ taking possibly any geometry (e.g., [2, 27, 44, 32]). A main concern on results in the form (4) is that the bound could be loose (i.e., a large $\max_{Q \in \mathcal{K}} \mathbb{E}_Q[\ell(f_{\text{dro}}^*, z)]$). This, in some sense, is unsurprising as the analysis only requires a confidence guarantee on the set $\mathcal{K}$, with no usage of other more specific properties, which is also the reason why the bound (4) is general. When $\mathcal{K}$ is suitably chosen, a series of work has shown that the bound (4) can achieve tightness in some well-defined sense. [48, 49] show that when $\mathcal{K}$ is a Kullback-Leibler divergence ball, $\max_{Q \in \mathcal{K}} \mathbb{E}_Q[\ell(f_{\text{dro}}^*, z)]$ in (4) is the minimal among all possible data-driven formulations that satisfy a given exponential decay rate on the confidence. [8, 50, 51] show that for divergence-based $\mathcal{K}$, $\max_{Q \in \mathcal{K}} \mathbb{E}_Q[\ell(f, z)]$ matches the confidence bound obtained from the standard central limit theorem by deducing that it is approximately $\mathbb{E}_{\hat{P}}[\ell(f, z)]$ plus a standard deviation term (see more related discussion momentarily).

Our result leverages part of the above "confidence translation" argument, but carefully twisted to obtain excess risk bounds for (1). We caution that (4) is a result on the validity of the estimated objective value $\max_{Q \in \mathcal{K}} \mathbb{E}_Q[\ell(f_{\text{dro}}^*, z)]$ in bounding the true objective value $\mathbb{E}_P[\ell(f_{\text{dro}}^*, z)]$. The excess risk (1), on the other hand, measures the generalization performance of a solution in comparison with the oracle best. The latter is arguably more intricate as it involves the unknown true optimal solution $f^*$ and, as we will see, (4) provides an intermediate building block in our analysis of (1).

## 2.2 Variability Regularization

In a series of work [30, 31, 7, 9, 51], it is shown that DRO using divergence-based ball, i.e., $\mathcal{K} = \{Q \in \mathcal{P} : D(Q, \hat{P}) \leq \eta\}$ for some threshold $\eta > 0$ and $D$ a $\phi$-divergence (e.g., Kullback-Leibler, $\chi^2$-distance), satisfies a Taylor-type expansion

$$\max_{Q \in \mathcal{K}} \mathbb{E}_Q[\ell(f, z)] = \mathbb{E}_{\hat{P}}[\ell(f, z)] + C_1(f)\sqrt{\eta} + C_2(f)\eta + \cdots \tag{5}$$

where $C_1(f)$ is the standard deviation of the loss function, $\sqrt{\text{Var}[\ell(f, z)]}$, multiplied by a constant that depends on $\phi$. Similarly, if $D$ is the Wasserstein distance and $\eta$ is of order $1/n$, (5) holds with $C_1(f)$ being the gradient norm or the Lipschitz norm [12, 10, 5, 52]. Furthermore, [17], which is perhaps closest to our work, studies MMD as the DRO statistical distance and derives a

high-probability bound for $\max_{Q \in \mathcal{K}} \mathbb{E}_Q[\ell(f,z)]$ similar to the RHS of (5), with $C_1(f)$ being the reproducing kernel Hilbert space (RKHS) norm of $\ell(f, \cdot)$. Results of the form (5) can be used to show that $\max_{Q \in \mathcal{K}} \mathbb{E}_Q[\ell(f,z)]$, upon choosing $\eta$ properly (of order $1/n$), gives a confidence bound on $\mathbb{E}_P[\ell(f,z)]$ [8, 51]. Moreover, this result can be viewed as a duality of the empirical likelihood theory [53, 12, 51, 54].

In connecting (5) to the solution performance, there are three implications. First, the robust objective function $\max_{Q \in \mathcal{K}} \mathbb{E}_Q[\ell(f,z)]$ can be interpreted as approximately a mean-variance optimization, and [7, 55] prove that, thanks to this approximation, the DRO solution can lower the variance of the attained loss which compensates for its under-performance in the expected loss, thus overall leading to a desirable risk profile. [7, 55] have taken a viewpoint that the variance of the attained loss is important in the generalization. On the other hand, when the expected loss, i.e., the true objective function $\mathbb{E}_P[\ell(f,z)]$, is the sole consideration, the approximation (5) is used in two ways. One way is to obtain bounds in the form

$$\mathbb{E}_P[\ell(f_{\text{dro}}^*, z)] \leq \min_{f \in \mathcal{F}} \{\mathbb{E}_P[\ell(f,z)] + C_1(f)/\sqrt{n}\} + O(1/n) \tag{6}$$

thus showing that DRO performs optimally, up to $O(1/n)$ error, on the variance-regularized objective function [9]. From this, [9] deduces that under special situations where there exists $f$ with both small risk $\mathbb{E}_P[\ell(f,z)]$ and variance $\text{Var}[\ell(f,z)]$, (6) can be translated into a small excess risk bound of order $1/n$. Moreover, such an order also desirably appears for DRO in some non-smooth problems where ERM could bear $1/\sqrt{n}$. The second way is to use DRO as a mathematical route to obtain uniform bounds in the form

$$\mathbb{E}_P[\ell(f,z)] \leq \mathbb{E}_{\hat{P}}[\ell(f,z)] + C_1(f)/\sqrt{n} + O(1/n), \; \forall f \in \mathcal{F}, \tag{7}$$

which is useful for proving the generalization of ERM. In particular, we can translate (7) into a high probability bound for $\mathbb{E}_P[\ell(\hat{f}^*, z)] - \mathbb{E}_P[\ell(f^*, z)]$ of order $1/\sqrt{n}$. This use is studied in, e.g., [56] in the case of Wasserstein and [17] in the case of MMD.

Despite these rich results, both (6) and (7), and their derived bounds on the excess risk (1), still depend on other candidates in the hypothesis class through the choice of the ball size $\eta$ or the coefficient $C_1(f)$. Our main message in this paper is that this dependence can be completely removed, when using the solution of a suitably constructed DRO. Note that this is different from localized results for ERM, e.g., local Rademacher complexities [57]. Although the latter establishes better generalization bounds by mitigating some dependence on the hypothesis class, local dependence on other candidates still exists and cannot be removed.

## 2.3 Risk Aversion

It is known that any coherent risk measure (e.g., conditional value-at-risk) [13, 14] of a random variable (in our case $\ell(f,z)$) is equivalent to the robust objective value $\max_{Q \in \mathcal{K}} \mathbb{E}_Q[\ell(f,z)]$ with a particular $\mathcal{K}$. Thus, DRO is equivalent to a risk measure minimization, which in turn explains its benefit in controlling tail performances. In machine learning, this rationale has been adopted to enhance performance on minority subpopulations and in safety or fairness-critical systems [58]. A related application is adversarial training in deep learning, in which RO or DRO is used to improve test performances on perturbed input data and adversarial examples [59, 60]. DRO has also been used to tackle distributional shifts in transfer learning [61, 62]. In these applications, numerical results suggest that RO and DRO can come at a degradation to the average-case performance [63, 60, 64] (though not universally, e.g., [65] observes that robust training can help reduce generalization error with very few training data). To connect, our result in this paper serves to justify a generalization performance of DRO even in the average case that can potentially outperform ERM.

## 3 General Results

We first present a general DRO bound.

**Theorem 3.1** (A General DRO Bound). *Let $\mathcal{Z} \subset \mathbb{R}^d$ be a sample space, $P$ be a distribution on $\mathcal{Z}$, $\{z_i\}_{i=1}^n$ be iid samples from $P$, and $\mathcal{F}$ be the function class. For loss function $\ell : \mathcal{F} \times \mathcal{Z} \to \mathbb{R}$, DRO solution $f_{dro}^* \in \arg\min_{f \in \mathcal{F}} \max_{Q \in \mathcal{K}} \mathbb{E}_Q[\ell(f,z)]$ satisfies that for any $\varepsilon > 0$,*

$$\mathbb{P}[\mathbb{E}_P \ell(f_{dro}^*, z) - \mathbb{E}_P \ell(f^*, z) > \varepsilon] \leq \mathbb{P}[P \notin \mathcal{K}] + \mathbb{P}[\max_{Q \in \mathcal{K}} \mathbb{E}_Q \ell(f^*, z) - \mathbb{E}_P \ell(f^*, z) > \varepsilon | P \in \mathcal{K}]. \tag{8}$$

*Proof of Theorem 3.1.* We first rewrite $\mathbb{E}_P[\ell(f_{\text{dro}}^*, z)] - \mathbb{E}_P[\ell(f^*, z)]$ as the following three terms: $[\mathbb{E}_P[\ell(f_{\text{dro}}^*, z)] - \max_{Q \in \mathcal{K}} \mathbb{E}_Q[\ell(f_{\text{dro}}^*, z)]] + [\max_{Q \in \mathcal{K}} \mathbb{E}_Q[\ell(f_{\text{dro}}^*, z)] - \max_{Q \in \mathcal{K}} \mathbb{E}_Q[\ell(f^*, z)]] + [\max_{Q \in \mathcal{K}} \mathbb{E}_Q[\ell(f^*, z)] - \mathbb{E}_P[\ell(f^*, z)]]$. Note that the second term $\max_{Q \in \mathcal{K}} \mathbb{E}_Q \ell(f_{\text{dro}}^*, z) - \max_{Q \in \mathcal{K}} \mathbb{E}_Q \ell(f^*, z) \leq 0$ almost surely by the DRO optimality of $f_{\text{dro}}^*$. Also, by definition of $\max_{Q \in \mathcal{K}} \mathbb{E}_Q[\cdot]$ as the worst-case objective function, we have $\max_{Q \in \mathcal{K}} \mathbb{E}_Q[\ell(f, z)] \geq \mathbb{E}_P[\ell(f, z)]$ for all $f \in \mathcal{F}$ as long as $P \in \mathcal{K}$. Thus,

$$\mathbb{P}[\mathbb{E}_P \ell(f_{\text{dro}}^*, z) - \mathbb{E}_P \ell(f^*, z) > \varepsilon] \leq \mathbb{P}[P \notin \mathcal{K}] + \mathbb{P}[\max_{Q \in \mathcal{K}} \mathbb{E}_Q \ell(f^*, z) - \mathbb{E}_P \ell(f^*, z) > \varepsilon, P \in \mathcal{K}]$$

$$\leq \mathbb{P}[P \notin \mathcal{K}] + \mathbb{P}[\max_{Q \in \mathcal{K}} \mathbb{E}_Q \ell(f^*, z) - \mathbb{E}_P \ell(f^*, z) > \varepsilon | P \in \mathcal{K}].$$

This completes our proof. $\square$

A key feature of the bound in Theorem 3.1 is that it requires only the true loss function $\ell(f^*, \cdot)$. This contrasts sharply with other bounds that rely on other candidates in the hypothesis class. To see where this distinction arises, note that in proof of Theorem 3.1, we divide $\mathbb{E}_P[\ell(f_{\text{dro}}^*, z)] - \mathbb{E}_P[\ell(f^*, z)]$ into three parts where the second term is trivially $\leq 0$ and the third term depends only on the true loss. The key is that the first term satisfies $\mathbb{E}_P[\ell(f_{\text{dro}}^*, z)] - \max_{Q \in \mathcal{K}} \mathbb{E}_Q[\ell(f_{\text{dro}}^*, z)] \leq 0$ as long as $P \in \mathcal{K}$, thanks to the worst-case definition of the robust objective function $\max_{Q \in \mathcal{K}} \mathbb{E}_Q[\ell(\cdot, z)]$. In contrast, in the ERM case for instance, the same line of analysis gives $[\mathbb{E}_P[\ell(\hat{f}^*, z)] - \mathbb{E}_{\hat{P}}[\ell(\hat{f}^*, z)]] + [\mathbb{E}_{\hat{P}}[\ell(\hat{f}^*, z)] - \mathbb{E}_{\hat{P}}[\ell(f^*, z)]] + [\mathbb{E}_{\hat{P}}[\ell(f^*, z)] - \mathbb{E}_P[\ell(f^*, z)]]$, and while the second and third terms are handled analogously, the first term depends on $\hat{f}^*$ that varies randomly among the hypothesis class and is typically handled by the uniform bound $\sup_{f \in \mathcal{F}} |\mathbb{E}_P[\ell(f, z)] - \mathbb{E}_{\hat{P}}[\ell(f, z)]|$ that requires empirical process analysis [66] and the complexity of the hypothesis class $\mathcal{F}$ or $\mathcal{L}$. Thus, in a sense, the worst-case nature of DRO "transfers" the uniformity requirement over the hypothesis class into alternate geometric requirements on only the true loss function.

## 4 Specialization to MMD DRO

### 4.1 Generalization Bounds

Our next step is to use Theorem 3.1 to derive generalization bounds for a concrete DRO formulation. In the following, we use ambiguity set $\mathcal{K}$ as a neighborhood ball of the empirical distribution measured by statistical distance, namely

$$\mathcal{K} = \{Q \in \mathcal{P} : D_{\text{MMD}}(Q, \hat{P}) \leq \eta\} \tag{9}$$

for a threshold $\eta > 0$. Moreover, we specialize in max mean discrepancy (MMD) [67] as the choice of distance $D_{\text{MMD}}(\cdot, \cdot)$. MMD is a distance derived from the RKHS norm, by using test functions constrained by this norm in an Integral Probability Metric (IPM) [68]. Specifically, let $k : \mathcal{Z} \times \mathcal{Z} \to \mathbb{R}$ be a positive definite kernel function on $\mathcal{Z}$ and $(\mathcal{H}, \langle \cdot, \cdot \rangle)$ be the corresponding RKHS. For any distributions $Q_1$ and $Q_2$, the MMD distance is defined by the maximum difference of the integrals over the unit ball $\{h \in \mathcal{H} : \|h\|_{\mathcal{H}} \leq 1\}$. That is, $D_{\text{MMD}}(Q_1, Q_2) := \sup_{h \in \mathcal{H}: \|h\|_{\mathcal{H}} \leq 1} \int h \mathrm{d}Q_1 - \int h \mathrm{d}Q_2$. In this way, MMD-DRO can be formulated as

$$\min_{f \in \mathcal{F}} \max_{Q: D_{\text{MMD}}(Q, \hat{P}) \leq \eta} \mathbb{E}_Q[\ell(f, z)] \tag{10}$$

MMD is known to be less prone to the curse of dimensionality [69–71], a property that we leverage in this paper, and such a property has allowed successful applications in statistical inference [67], generative models [72, 73], reinforcement learning [74], and DRO [17, 75]. In particular, [76] studies the duality and optimization procedure for MMD DRO. For a recent comprehensive review of MMD, we refer readers to [77].

In the sequel, we adopt bounded kernels, i.e., $\sup_{z,z' \in \mathcal{Z}} \sqrt{k(z, z')} < +\infty$, to conduct our analysis. This assumption applies to many popular choices of kernel functions and more importantly, guarantees the so-called kernel mean embedding (KME) [77] is well-defined in RKHS. That is, $\mu_Q := \int k(z, \cdot) \mathrm{d}Q \in \mathcal{H}, \ \forall \ Q \in \mathcal{P}$. This well-definedness of KME gives two important implications. One is that MMD is equivalent to the norm distance in KME: $D_{\text{MMD}}(P, Q) = \|\mu_P - \mu_Q\|_{\mathcal{H}}$ [78, 67]. Second, it bridges the expectation and the inner product in the RKHS space [70]: $\forall \ f \in \mathcal{H}$, $\mathbb{E}_{x \sim P}[f(x)] = \int_x \langle f, k(x, \cdot) \rangle \mathrm{d}P(x) = \langle f, \mu_P \rangle$. Both properties above will facilitate our analysis. We refer the reader to Appendix A for further details on KME.

**Theorem 4.1** (Generalization Bounds for MMD DRO). *Adopt the notation and assumptions in Theorem 3.1. Let $\mathcal{Z}$ be a compact subspace of $\mathbb{R}^d$, $k : \mathcal{Z} \times \mathcal{Z} \to \mathbb{R}$ be a bounded continuous positive definite kernel, and $(\mathcal{H}, \langle \cdot, \cdot \rangle_{\mathcal{H}})$ be the corresponding RKHS. Suppose also that (i) $\sup_{z \in \mathcal{Z}} \sqrt{k(z,z)} \leq K$; (ii) $\ell(f^*, \cdot) \in \mathcal{H}$ with $\|\ell(f^*, \cdot)\|_{\mathcal{H}} \leq M$. Then, for all $\delta \geq 0$, if we choose ball size $\eta = \frac{K}{\sqrt{n}}(1 + \sqrt{2\log(1/\delta)})$ for MMD DRO* (10), *then MMD DRO solution $f^*_{M\text{-}dro}$ satisfies*

$$\mathbb{E}_P[\ell(f^*_{M\text{-}dro}, z)] - \mathbb{E}_P[\ell(f^*, z)] \leq \frac{2KM}{\sqrt{n}}(1 + \sqrt{2\log(1/\delta)})$$

*with probability at least $1 - \delta$.*

We briefly overview our proof of Theorem 4.1, leaving the details to Appendix B. In view of Theorem 3.1, the key lies in choosing the ball size $\eta$ to make a good trade-off between $\mathbb{P}[D_{\mathrm{MMD}}(P, \hat{P}) \geq \eta]$ and an upper bound of $\max_{Q \in \mathcal{K}} \mathbb{E}_Q[\ell(f^*, z)] - \mathbb{E}_P[\ell(f^*, z)]$ when $D_{\mathrm{MMD}}(P, \hat{P}) \leq \eta$. The former can be characterized by the established concentration rate of KME (See Proposition B.1). To get the latter bound, we use the properties of KME to translate expectations into inner products:

$$\max_{Q \in \mathcal{K}} \mathbb{E}_Q[\ell(f^*, z)] - \mathbb{E}_P[\ell(f^*, z)] = \max_{Q \in \mathcal{K}} \langle \ell(f^*, \cdot), \mu_Q - \mu_P \rangle \leq \max_{Q \in \mathcal{K}} \|\ell(f^*, \cdot)\|_{\mathcal{H}} D_{\mathrm{MMD}}(Q, P)$$

$$\leq \|\ell(f^*, \cdot)\|_{\mathcal{H}} \max_{Q \in \mathcal{K}} \{D_{\mathrm{MMD}}(Q, \hat{P}) + D_{\mathrm{MMD}}(\hat{P}, P)\} \leq \|\ell(f^*, \cdot)\|_{\mathcal{H}} \cdot 2\eta. \qquad (11)$$

From these, we can choose $\eta$ as indicated in Theorem 4.1 to conclude the result. In a nutshell, here we harness the compatibility between $\mathcal{K}$ and the true loss function $\ell(f^*, \cdot)$, more specifically the RKHS norm on $\ell(f^*, \cdot)$, and the dimension-free concentration of KME. Our general bound in Theorem 3.1 allows us to stitch these two ingredients together to obtain generalization bounds for MMD DRO.

## 4.2 Strengths and Limitations Compared with ERM

Our generalization bound in Theorem 4.1 in fact still utilizes a uniformity argument, hidden in the distance construction. Note that our bound comprises essentially the product of $\|\ell(f^*, \cdot)\|_{\mathcal{H}}$, which depends on the true loss function, and $\eta$, the size of the ambiguity set. The latter is controlled by $D_{\mathrm{MMD}}(P, \hat{P})$, which is equivalent to the supremum of the empirical process indexed by $\mathcal{H}_1 := \{h \mid h \in \mathcal{H}, \|h\|_{\mathcal{H}} \leq 1\}$. Note that this supremum is expressed through $\sup_{z \in \mathcal{Z}} \sqrt{k(z,z)}$, which is independent of the hypothesis class.

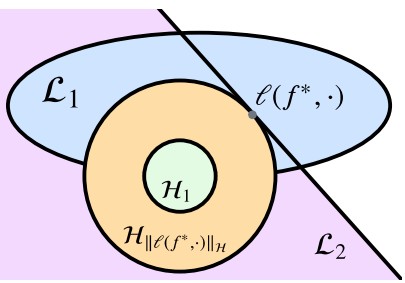

Figure 1: Let $\mathcal{L}_1$ and $\mathcal{L}_2$ be two hypothesis classes with the same true loss function $\ell(f^*, \cdot)$. $\mathcal{L}_1$ corresponds to functions in the oval and $\mathcal{L}_2$ corresponds to functions below the straight line. In ERM, the uniform bound is taken directly over $\mathcal{L}_1$ (blue area) or $\mathcal{L}_2$ (pink area), while DRO gives the same bound that corresponds to uniformity over $\mathcal{H}_{\|\ell(f^*, \cdot)\|_{\mathcal{H}}}$ (orange area).

To understand this further, Figure 1 provides geometric representations on the classes of functions over which ERM and our DRO apply uniform bounds. From this we also argue how DRO relies less heavily on the hypothesis class. In ERM, the uniform bound is taken directly over the hypothesis class, say $\mathcal{L}$. In DRO, from our discussion above, the uniform bound is taken over a dilation of the $\mathcal{H}_1$-ball by a factor $\|\ell(f^*, \cdot)\|_{\mathcal{H}}$, equivalently $\mathcal{H}_{\|\ell(f^*, \cdot)\|_{\mathcal{H}}} := \{h \mid h \in \mathcal{H}, \|h\|_{\mathcal{H}} \leq \|\ell(f^*, \cdot)\|_{\mathcal{H}}\}$. Thus, if two hypothesis classes contain the same true loss function $\ell(f^*, \cdot)$, their DRO generalization bounds would be the same, while ERM could differ. In this sense, DRO generalization exhibits less reliance on the hypothesis class.

With the above understanding, we can see several situations where DRO could outperform ERM: (i) *When $\|\ell(f^*, \cdot)\|_{\mathcal{H}}$ is small compared to the size of $\mathcal{L}$.* Though one may argue that $\ell(f^*, \cdot)$ is typically unknown and hence we ultimately resort to using uniformity over $\mathcal{L}$ to obtain any computable bound, our bound in terms of $\ell(f^*, \cdot)$ explains why DRO can sometimes substantially outperform ERM. In Section 6, we provide an example where $\|l(f^*, \cdot)\|_{\mathcal{H}} = 0$ and the excess risk of DRO achieves zero with high probability. (ii) *When uniformly bounding over $\mathcal{H}_1$ is "cheap".* Many commonly used kernel spaces are bounded and therefore uniform bounds over $\mathcal{H}_1$ are trivial to establish (by Proposition B.1), which in turn leads to generalization bounds that depend only on a

dilation (by $\|\ell(f^*, \cdot)\|_{\mathcal{H}}$) of these uniform bounds over $\mathcal{H}_1$. This is regardless of how complex the hypothesis class $\mathcal{L}$ is, including when $\|\mathcal{L}\|_{\mathcal{H}} = \infty$ or $\mathcal{L} = \mathcal{H}$. ERM, in contrast, uses uniformity over $\mathcal{L}$ and can result in an exploding generalization bound when $\|\mathcal{L}\|_{\mathcal{H}} = \infty$. (iii) *When $\mathcal{L}$ resembles the geometry of $\mathcal{H}$.* For instance, when $\mathcal{L} = \mathcal{H}_B := \{h \mid h \in \mathcal{H}, \|h\|_{\mathcal{H}} \leq B\}$, the ERM bound yields $2B \sup_{h \in \mathcal{H}_1} \{Ph - \hat{P}h\}$ since $B = \sup_{\ell \in \mathcal{L}} \|\ell(\cdot)\|_{\mathcal{H}}$, while the DRO bound gives $2\|\ell(f^*, \cdot)\|_{\mathcal{H}} \sup_{h \in \mathcal{H}_1} \{Ph - \hat{P}h\}$ that depends only on the true loss function and is smaller.

Despite the strengths, our bound in Theorem 4.1 also reveals situations where DRO could under-perform ERM: When $\mathcal{L}$ has a shape deviating from the RKHS ball such that $\mathcal{L} \subset \mathcal{H}_{\|\ell(f^*, \cdot)\|_{\mathcal{H}}}$, then taking uniform bounds over $\mathcal{H}_{\|\ell(f^*, \cdot)\|_{\mathcal{H}}}$ is costly compared to over $\mathcal{L}$, in which case DRO generalization bound is worse than ERM. Moreover, note that $\ell(f^*, \cdot)$, the optimal loss within $\mathcal{L}$, can shift when $\mathcal{L}$ becomes richer. In this sense Theorem 4.1 cannot completely remove the dependency on the hypothesis class, but rather diminishes it to the minimum possible: The resulting bound does not rely on any other candidates in the hypothesis class except $\ell(f^*, \cdot)$.

*Remark* 4.2. There exists another line of work that establishes generalization bounds with a faster rate $O(n^{-1})$ for ERM [79, 57, 80, 81] and $\phi$-divergence DRO [9]. However, they are all based on additional curvature conditions on $\mathcal{F}$ or $\mathcal{L}$ to relate the loss/risk and variance. Improving our bounds towards $O(n^{-1})$ rate is potentially achievable and constitutes future work.

## 4.3 Comparison with [17]

For bounds that do not utilize conventional complexity measures, to the best of our knowledge, they are still dependent on other candidates in the hypothesis class; recall our discussion in Section 1. As a concrete example, we compare our results with [17], which also studies generalization related to MMD DRO and appears the most relevant to our paper. First, different from our research goal, [17] aims to use MMD DRO as an analysis tool to derive uniform bounds for $\mathbb{E}_P[\ell(f, z)] - \mathbb{E}_{\hat{P}}[\ell(f, z)], \forall f \in \mathcal{F}$. More specifically, the main argument for their Theorem 4.2 is that, with high probability, one has

$$\mathbb{E}_P[\ell(f, z)] - \mathbb{E}_{\hat{P}}[\ell(f, z)] = O(\|\ell(f, \cdot)\|_{\mathcal{H}} \sqrt{\tfrac{\log(1/\delta)}{n}})$$ for any given $f$. Although this bound seems to depend on a single loss function $\ell(f, \cdot)$, any algorithmic output, say $f_{\text{data}}$, needs to be learnt from data and is thus random. This means that to construct a generalization bound, one has to either average over $f_{\text{data}}$ or take a supremem over $f \in \mathcal{F}$. [17] adopts the latter approach and establishes a bound

$$\mathbb{E}_P[\ell(f, z)] - \mathbb{E}_{\hat{P}}[\ell(f, z)] = O(\sup_{f \in \mathcal{F}} \|\ell(f, \cdot)\|_{\mathcal{H}} \sqrt{\tfrac{\log(1/\delta)}{n}}), \ \forall \ f \in \mathcal{F}$$ in their Theorem 4.2. Our Theorem 4.1, in contrast, only depends on $\|\ell(f^*, \cdot)\|_{\mathcal{H}}$ which is potentially much tighter and provides a distinct perspective.

Second, our Theorem 4.1 also distinguishes from [17] in that our bound applies specifically to the MMD DRO solution $f^*_{\text{M-dro}}$ and reveals a unique property of this solution. This should be contrasted to [17] whose result is a uniform bound for any $f \in \mathcal{F}$, without any specialization to the DRO solution. This also relates to our next discussion.

## 4.4 Comparison with RKHS Regularized ERM

We also compare our results to RKHS regularized ERM:

$$\min_{f \in \mathcal{F}} \ell(f, z) + \eta \|f\|_{\mathcal{H}}. \tag{12}$$

When we additionally assume $\mathcal{L} \subset \mathcal{H}$, it is known that MMD DRO is equivalent to

$$\min_{f \in \mathcal{F}} \ell(f, z) + \eta \|\ell(f, \cdot)\|_{\mathcal{H}} \tag{13}$$

according to [17, Theorem 3.1]. Although (12) and (13) appear similar, their rationales are different. When regularizing $\|f\|_{\mathcal{H}}$, the goal is to maintain the simplicity of $f$. In comparison, through (13) we are in disguise conducting DRO and utilizing its variability regularization property (Recall Section 2.2). This observation motivates [17] to consider regularizing $\ell(f, \cdot)$ instead of $f$ itself. Our Theorem 4.1 reveals the benefit of regularizing $\ell(f, \cdot)$ at a new level: Regularizing $\|\ell(f, \cdot)\|_{\mathcal{H}}$ increases the likelihood of $\mathbb{E}_P[\ell(f^*_{\text{M-dro}}, z)] - \max_{Q \in \mathcal{K}} \mathbb{E}_Q[\ell(f^*_{\text{M-dro}}, z)] \leq 0$, which is argued by looking at the *distribution* space, instead of the loss function space, more specifically from the likelihood of a DRO ambiguity set covering $P$. Because of this, we can establish generalization bounds that depend on the hypothesis class only through the true loss $\ell(f^*, \cdot)$ for RKHS-regularized

ERM when the regularization is on $\ell(f, \cdot)$. Furthermore, thanks to this we can relax the assumption $\mathcal{L} \subset \mathcal{H}$ adopted in (13) to merely $\ell(f^*, \cdot) \in \mathcal{H}$ which is all we need in MMD DRO. Lastly, we point out that our analysis framework is not confined only to MMD, but can be applied to other distances in defining DRO (See Section 5).

### 4.5 Extension to $\ell(f^*, \cdot) \notin \mathcal{H}$

In Theorem 4.1, we assume (i) $\sup_{x \in \mathcal{X}} \sqrt{k(x, x)} \leq K$; (ii) $\ell(f^*, \cdot) \in \mathcal{H}$ with $\|\ell(f^*, \cdot)\|_{\mathcal{H}} \leq M$. Although (i) is natural for many kernels, (ii) can be restrictive since many popular loss functions do not satisfy $\ell(f^*, \cdot) \in \mathcal{H}$. For instance, if $k$ is a Gaussian kernel defined on $\mathcal{Z} \subset \mathbb{R}^d$ that has nonempty interior, then any nonzero polynomial on $\mathcal{Z}$ does not belong to the induced RKHS $\mathcal{H}$ [82]. This is in fact a common and well-known problem for almost all kernel methods [83], though many theoretical kernel method papers just assume the concerned function $g \in \mathcal{H}$ for simplicity, e.g., [84, 85, 17]. This theory-practice gap motivates us to extend our result to $\ell(f^*, \cdot) \notin \mathcal{H}$.

To this end, note that under universal kernels [86], any bounded continuous function can be approximated arbitrarily well (in $L^\infty$ norm) by functions in $\mathcal{H}$. However, the RKHS norm of the approximating function, which appears in the final generalization bound, may grow arbitrarily large as the approximation becomes finer [87, 83]. This therefore requires analyzing a trade-off between the function approximation error and RKHS norm magnitude. More specifically, we define the approximation rate $I(\ell(f^*, \cdot), R) := \inf_{\|g\|_{\mathcal{H}} \leq R} \|\ell(f^*, \cdot) - g\|_\infty$, and $g_R := \arg \inf_{\|g\|_{\mathcal{H}} \leq R} \|\ell(f^*, \cdot) - g\|_\infty$.[1] As a concrete example, we adopt the Gaussian kernel and the Sobolev space to quantify such a rate and establish Theorem 4.3 below.

**Theorem 4.3** (Generalization Bounds for Sobolev Loss Functions). *Let $(\mathcal{H}, \| \cdot \|_{\mathcal{H}})$ be the RKHS induced by $k(z, z') = \exp(-\|z - z'\|_2^2/\sigma^2)$ on $\mathcal{Z} = [0, 1]^d$. Suppose that $\ell(f^*, \cdot)$ is in the Sobolev space $H^d(\mathbb{R}^d)$. Then, there exists a constant $C$ independent of $R$ but dependent on $\ell(f^*, \cdot)$, $d$, and $\sigma$ such that for all $\delta \geq 0$, if we choose ball size $\eta = \frac{1}{\sqrt{n}}(1 + \sqrt{2 \log(1/\delta)})$ for MMD DRO* (10)*, then MMD DRO solution $f^*_{M\text{-}dro}$ satisfies*

$$\mathbb{E}_P[\ell(f^*_{M\text{-}dro}, z)] - \mathbb{E}_P[\ell(f^*, z)] \leq 2 \inf_{R \geq 1} \left\{ C(\log R)^{-\frac{d}{16}} + \frac{R}{\sqrt{n}}(1 + \sqrt{2 \log(1/\delta)}) \right\} \qquad (14)$$

*with probability at least $1 - \delta$.*

We briefly outline our proof, leaving the details to Appendix C. When $\ell(f^*, \cdot) \notin \mathcal{H}$, we cannot use KME to rewrite expectation as inner products as shown in (11). Rather, we will approximate $\ell(f^*, \cdot)$ by some $g_R \in \mathcal{H}$ and obtain $\mathbb{E}_P[\ell(f^*, z)] \leq \mathbb{E}_P[g_R(z)] + I(\ell(f^*, \cdot), R)$. Since $g_R \in \mathcal{H}$, similar to (11) we can derive a generalization bound of $g_R$, which appears in the second term of (14). Then, it suffices to take infinum over $R \geq 1$ to establish the aforementioned trade-off between the approximation error $I(\ell(f^*, \cdot), R)$ (the first term of (14)) and the RKHS norm $\|g_R\|_{\mathcal{H}} \leq R$ (the second term of (14)).

Compared to Theorem 4.1, here we have to pay an extra price on the solution quality when the loss function is less compatible with the chosen RKHS, i.e., $\ell(f^*, \cdot) \notin \mathcal{H}$. Concretely, in the setting of Theorem 4.3, $I(\ell(f^*, \cdot), R) \leq C(\log R)^{-d/16}$ [87] and this results in a worse convergence rate in $n$. For example, inserting $R = \sqrt{n}(\log n)^{-d/16}$ into (14) yields $\mathcal{O}(\max\{C, \sqrt{\log(1/\delta)}\}(\log n)^{-16/d})$. This convergence rate is rather slow. However, it mainly results from the slow approximation rate $I(\ell(f^*, \cdot), R)$ established by non-parametric analysis. If we further utilize the structure of $\ell(f^*, \cdot)$, then a faster convergence rate or even close to $O(n^{-1/2})$ rate can be potentially achieved. Nonetheless, our main message in Theorem 4.3 is that even if $\ell(f^*, \cdot) \notin \mathcal{H}$, our bound (14) still depends minimally on the hypothesis class $\mathcal{L}$: The bound depends solely on the true loss function $\ell(f^*, \cdot)$, independent of any other $\ell(f, \cdot) \in \mathcal{L}$.

## 5 Specialization to 1-Wasserstein and $\chi^2$-divergence DRO

Our framework can also be specialized to DROs using other statistical distances. Below, we derive results for DROs based on the 1-Wasserstein distance and $\chi^2$-divergence, and discuss their implica-

---

[1]$g_R$ is well-defined since for fixed $R$, $h(g) = \|\ell(f^*, \cdot) - g\|_\infty$ is a continuous function over a compact set $\{g : g \in \mathcal{H}, \|g\|_{\mathcal{H}} \leq R\}$.

tions and comparisons with MMD DRO. Due to page limit, we delegate more detailed discussions to Appendix D. Let $f^*_{\text{W-dro}}$ be the solution to 1-Wasserstein DRO (See Appendix D.1) and $f^*_{\chi^2\text{-dro}}$ be the solution to $\chi^2$-divergence DRO (See Appendix D.2). Also denote $\|\cdot\|_{\text{Lip}}$ as the Lipschitz norm. Then, we have the following generalization bounds.

**Theorem 5.1** (Generalization Bounds for 1-Wasserstein DRO). *Suppose that $P$ is a light-tailed distribution in the sense that there exists $a > 1$ such that $A := \mathbb{E}_P[\exp(\|z\|^a)] < \infty$. Then, there exists constants $c_1$ and $c_2$ that only depends on $a, A$, and $d$ such that for any given $0 < \delta < 1$, if we choose ball size $\eta = \left(\frac{\log(c_1/\delta)}{c_2 n}\right)^{1/\max\{d,2\}}$ and $n \geq \frac{\log(c_1/\delta)}{c_2}$, then $f^*_{\text{W-dro}}$ satisfies $\mathbb{E}_P[\ell(f^*_{\text{W-dro}}, z)] - \mathbb{E}_P[\ell(f^*, z)] \leq 2\|\ell(f^*, \cdot)\|_{\text{Lip}}(\frac{\log(c_1/\delta)}{c_2 n})^{1/\max\{d,2\}}$ with probability at least $1 - \delta$.*

**Theorem 5.2** (Generalization Bounds for $\chi^2$-divergence DRO on Discrete Distributions). *Suppose that $P$ is a discrete distribution with $m$ support with $\mathbb{P}[z = z_i] = p_i$. Suppose that (i) $p_{\min} = \min_{1 \leq i \leq m} p_i > 0$; (ii) $\|\ell(f^*, \cdot)\|_\infty < +\infty$. Then, for all $0 < \delta < 1$, if we choose ball size $\eta = \frac{1}{n}(m + 2\log(4/\delta) + 2\sqrt{m\log(4/\delta)})$ for $\chi^2$-divergence DRO, then for all $n \geq \frac{10^6 m^2}{p_{\min}^3 \delta^2}$, $f^*_{\chi^2\text{-dro}}$ satisfies $\mathbb{E}_P[\ell(f^*_{\chi^2\text{-dro}}, z)] - \mathbb{E}_P[\ell(f^*, z)] \leq \|\ell(f^*, \cdot)\|_\infty \{\frac{1}{\sqrt{n}}\sqrt{m + 2\log(4/\delta) + 2\sqrt{m\log(4/\delta)}} + \sqrt{\frac{2\log(2/\delta)}{n}}\}$ with probability at least $1 - \delta$.*

Like Theorem 4.1, Theorems 5.1 and 5.2 are obtained by analyzing the two components in Theorem 3.1, which correspond to: (1) the ambiguity set contains true distribution $P$ with high confidence and (2) compatibility of the statistical distance with $\ell(f^*, \cdot)$. The bounds in Theorems 5.1 and 5.2 involve only $\ell(f^*, \cdot)$ but not other candidates in the hypothesis class. Notice that if $P$ is continuous, then using $\chi^2$-divergence, and more generally $\phi$-divergence (See Definition D.3), for DRO entails a bit more intricacies than MMD and Wasserstein DRO. Due to the absolute continuity requirement in the definition, if $P$ is continuous, then $\phi$-divergence ball centered at $\hat{P}$ will never cover the truth, which violates the conditions required in developing our generalization bounds. In Theorem 5.2, to showcase how generalization bounds of $\phi$-divergence DRO depend on the hypothesis class, we present an explicit bound for discrete $P$ and $\chi^2$-divergence, for which we can use $\hat{P}$-centered ambiguity set. To remedy for continuous $P$, one can set the ball center using kernel density estimate or a parametric distribution, albeit with the price of additional approximation errors coming from the "smoothing" of the empirical distribution. In Appendix D.2, we present a general bound (Theorem D.4) for $\phi$-divergence DRO that leaves open the choice of the ball center and the function $\phi$. Once again, this generalization bound involves only $\ell(f^*, \cdot)$ but not other candidates in the hypothesis class. However, we would leave the explicit bounds of other cases than that of Theorem 5.2, including the choice and usage of the remedial smoothing distributions in the continuous case, to separate future work.

Moreover, in our result for Wasserstein DRO, the rates in terms of $n$ is $n^{-1/\max\{d,2\}}$, which is slower than our result for MMD DRO in Theorem 4.1 and the classical ERM rate $n^{-1/2}$. The slow rate of Wasserstein DRO comes from the large ball size needed to confidently cover the true $P$ and control the first term of (8). Specifically, the ball size is controlled by the 1-Wasserstein distance $D_{W_1}(P, \hat{P})$, which is equivalent to the supremum of the empirical process indexed by $\text{Lip}_1 := \{h \mid \|h\|_{\text{Lip}} \leq 1\}$ by the dual representation of $D_{W_1}(\cdot, \cdot)$ [88]. Similar to our discussion in Section 4.2, Theorem 5.1 shows a tradeoff between the hypothesis class dependence and the rate in $n$ that differs from ERM. More precisely, Wasserstein DRO has a generalization that favorably relies only on the true loss

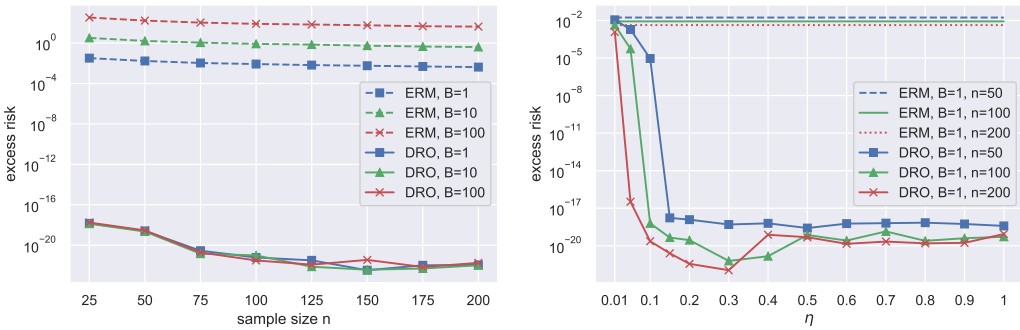

(a) Performance of ERM and DRO under varying $n$   (b) Performance of ERM and DRO under varying $\eta$

Figure 2: Excess risks of ERM and DRO

$\ell(f^*, \cdot)$, but scales inferiorly in terms of $n$ (due to uniformity over $\mathrm{Lip}_1$). It is thus most suited when the complexity of hypothesis class outweighs the convergence rate consideration. In Appendix E, we will describe the distribution shift setting where such a tradeoff can be amplified.

We also note that, unlike Wasserstein DRO, $\chi^2$-divergence DRO retains a rate of $1/\sqrt{n}$. However, this is because Theorem 5.2 has focused on the finite-support case, where the number of support points $m$ comes into play in the multiplicative factor instead of the rate. If we consider continuous $P$ and use a smooth baseline distribution (instead of $\hat{P}$), we expect the rate of $\chi^2$-divergence DRO to deteriorate much like the case of Wasserstein DRO and there may exist a similar tradeoff.

## 6 Numerical Experiments

We conduct simple experiments to study the numerical behaviors of our MMD DRO and compare with ERM on simulated data, which serves to illustrate the potential of MMD DRO in improving generalization and validate our developed theory. We adapt the experiment setups from [9, Section 5.2] and consider a quadratic loss with linear perturbation: $l(\theta, z) = \frac{1}{2}\|\theta - v\|_2^2 + z^\top(\theta - v)$, where $z \sim \mathrm{Unif}[-B, B]^d$ with constant $B$ varying from $\{1, 10, 100\}$ in the experiment. Here, $\mathcal{F} = \{\theta \mid \theta \in \Theta\}$ and we use $f$ and $\theta$ interchangeably. $v$ is chosen uniformly from $[1/2, 1]^d$ and is known to the learner in advance. In applying MMD DRO, we use Gaussian kernel $k(z, z') = \exp(-\|z - z'\|_2^2/\sigma^2)$ with $\sigma$ set to the median of $\{\|z_i - z_j\|_2 \mid \forall\, i, j\}$ according to the well-known median heuristic [67]. To solve MMD DRO, we adopt the semi-infinite dual program and the constraint sampling approach from [76], where we uniformly sample a constraint set of size equal to the data size $n$. The sampled program is then solved by CVX [89] and MOSEK [90]. Each experiment takes the average performance of 500 independent trials. Our computational environment is a Mac mini with Apple M1 chip, 8 GB RAM and all algorithms are implemented in Python 3.8.3.

In Figure 2(a), we present the excess risks of ERM and DRO, i.e., $\mathbb{E}_P[\ell(f_{\mathrm{data}}, z)] - \mathbb{E}_P[\ell(f^*, z)]$, where $f_{\mathrm{data}}$ denotes the ERM or DRO solution. We set $d = 5$ and tune the ball size via the best among $\eta \in \{0.01, 0.05, 0.1, 0.15, 0.2, 0.3, \ldots, 1.0\}$ (more details to be discussed on Figure 2(b) momentarily). Figure 2(a) shows that, as $n$ increases, both DRO and ERM incur a decreasing loss. More importantly, DRO appears to perform much better than ERM. For $n = 200$ and $B = 100$ for instance, the expected loss of DRO is less than $10^{-20}$, yet that of ERM remains at around $10^1$.

We attribute the outperformance of DRO to its dependency on the hypothesis class only through the true loss function $\ell(f^*, \cdot)$. This can be supported by an illustration on the effect of ball size $\eta$. In Figure 2(b), we present excess risk $\mathbb{E}_P[\ell(f_{\mathrm{data}}, z)] - \mathbb{E}_P[\ell(f^*, z)]$ for $B = 1, d = 5, n = 50$, and varying $\eta$. We see that the excess risk of the DRO solution drops sharply when $\eta$ is small, and then for sufficiently big $\eta$, i.e., $\eta \in [0.5, 1]$, it remains at a fixed level of $10^{-19}$, which is close to the machine accuracy of zero (such an "L"-shape excess loss also occurs under other choices of sample size $n$, and we leave out those results to avoid redundancy). Such a phenomenon coincides with our theorems. First, note that $f^* = v$ and the optimal loss function satisfies $\|\ell(f^*, \cdot)\|_{\mathcal{H}} = \|0\|_{\mathcal{H}} = 0$, so that our assumptions in Theorem 4.1 are satisfied. Recall that by Theorem 3.1,

$$\mathbb{P}[\mathbb{E}_P\ell(f^*_{\mathrm{M\text{-}dro}}, z) - \mathbb{E}_P\ell(f^*, z) > \varepsilon] \leq \mathbb{P}[P \notin \mathcal{K}] + \mathbb{P}[\max_{Q \in \mathcal{K}} \mathbb{E}_Q\ell(f^*, z) - \mathbb{E}_P\ell(f^*, z) > \varepsilon \mid P \in \mathcal{K}].$$

Conditioning on $P \in \mathcal{K}$, we have $f^*_{\mathrm{M\text{-}dro}} = f^* = v$ and $\max_{Q \in \mathcal{K}} \mathbb{E}_Q[\ell(f^*_{\mathrm{M\text{-}dro}}, z)] = \mathbb{E}_P[\ell(f^*, z)] = 0$. Thus, the above inequality yields $\mathbb{P}[\mathbb{E}_P[\ell(f^*_{\mathrm{M\text{-}dro}}, z)] - \mathbb{E}_P[\ell(f^*, z)] > \varepsilon] \leq \mathbb{P}[P \notin \mathcal{K}]$. The proof of Theorem 4.1 reveals that $\forall\, \delta > 0$, if the ball size is big enough, namely $\eta \geq \frac{K}{\sqrt{n}}(1 + \sqrt{2\log(1/\delta)})$, then excess risk will vanish, i.e., $\mathbb{E}_P[\ell(f^*_{\mathrm{M\text{-}dro}}, z)] - \mathbb{E}_P[\ell(f^*, z)] = 0$ with probability $\geq 1 - \delta$. This behavior shows up precisely in Figure 2(b).

Our discussion above is not to suggest the choice of $\eta$ in practice. Rather, it serves as a conceptual proof for the correctness of our theorem. In fact, to the best of our knowledge, none of the previous results, including the multiple approaches to explain the generalization of DRO reviewed in Section 2, can explain the strong outperformance of DRO with the "L"-shape phenomenon in this example. These previous results focus on either absolute bounds (Section 2.1) or using variability regularization (Section 2.2) whose bounds, when converted to excess risks, typically increase in $\eta$ when $\eta$ is sufficiently large and, moreover, involve complexity measures on the hypothesis class (e.g., the $\phi$-divergence case in [9, Theorems 3, 6] and the Wasserstein case in [56, Corollaries 2, 4]). Our developments in Section 3 thus appear to provide a unique explanation for the significant outperformance of DRO in this example.

## Acknowledgments and Disclosure of Funding

We gratefully acknowledge support from the National Science Foundation under grants CAREER CMMI-1834710 and IIS-1849280. We also thank Nathan Kallus and Jia-Jie Zhu for the greatly helpful discussion that helps improve this paper.

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
