# OpenReview forum: "Generalization Bounds with Minimal Dependency on Hypothesis Class via Distributionally Robust Optimization"
_NeurIPS.cc/2022/Conference — NeurIPS 2022 Accept_

### Official Review · Reviewer_xNyC · 2022-07-10

**Rating:** 7
**Confidence:** 5
**Soundness:** 3 good
**Presentation:** 3 good
**Contribution:** 3 good

**Summary:**

The authors present a DRO framework to derive generalization bounds, where the bounds crucially depend on the chosen ambiguity set, but not on the complexity of the hypothesis class, as opposed to existing ERM-type generalization bounds. The authors then exemplify the presented genearlization bounds for the maximum mean discrepancy (MMD) ambiguity set.

**Questions:**

1) In Theorem 4.1, how can you practically verify the assumption \|\ell(f^*,.)\|_H \leq M? Since you don’t know f^*, I guess you require a uniform bound of the form \sup_{f\in H} \ell(f,.)? How would you practically estimate M?
2) Do you have any idea how tight the bound in Theorem 4.1 is? Do there exist lower bounds?
3) Let us assume to be in the setting of Theorem 5.1. How do you compute f^*_DRO? Is there a tractable reformulation? Is that similar to [18]?

In general the paper is well written and easy to follow. I just have the following (minor) suggestions to improve the presentation:
1) I don’t see the benefit of introducing the non-standard notation of P\ell(f,.) for E_P[\ell(f,z)]. I find this notation mathematically imprecise (since P is a probability measure and therefore cannot act on functions, unless you overload notation. Moreover, I find this notation confusing and would prefer to use the expected value notation.
2) Please correct the following typos:
- line 121, “bals
- line 290, “supremum
3) As you mention distributional shifts in transfer learning, there is a recent DRO-related work on that topic you might want to compare your work against (Sutter, Krause, Kuhn, NeurIPS 2021, Robust Generalization despite Distribution Shift via Minimum Discriminating Information)
4) It would be nice to remind the reviewer about the definition of a Sobolev space (required to understand Theorem 5.1)

**Limitations:**

I see the “usual” DRO limitations also in this work, which are: (i) assumption of iid data, (ii) subjectivity of which ambiguity set to choose and (iii) need of a tractable reformulation to solve the DRO problem.

There are, however, good answers in the DRO literature to all three limitations, so I assume the presented framework in subsequent works can address these limitations.


**Strengths And Weaknesses:**

Strengths: I agree with the authors that most existing DRO works derive upper-confidence-type results where a data-driven decision f_data and a corresponding upper-confidence bound $J_{data}$ are generated with the corresponding statistical certificate $P[ E[\ell(f_{data},z)] <= J_{data} ]>= 1-\beta$. This work presents “true” generalization bounds quantifying the excess risk. Therefore, this approach clearly is novel and of high interested to a learning community.

Weaknesses: I think in the presented framework, there could me more properties being explored. For example, can you derive explicit generalization bounds for ambiguity sets other than MMD, e.g., f-divergence balls, moment-ambiguity sets? Moreover, how would a decision maker decide which ambiguity set to choose?


Post-Revision Comment:
I would like to thank the authors for the careful revision and for providing convincing explanations to all my questions. I will raise my rating to 7 and recommend acceptance of this paper.

---

> ### Author Response · Authors · 2022-08-02
> **Response to Reviewer xNyC**
>
> We thank the reviewer for the careful reading and helpful feedback, and are also glad to hear the positive opinion of the reviewer. The reviewer's main concern on our weakness regards more extensive investigations on other types of DRO. To respond to this (as well as similar comments from another reviewer), we have added a substantial expansion of our theory, now in Appendices D, E and F in our revised paper, to cover:
>
> 1) 1-Wasserstein DRO;
>
> 2) $\phi$-divergence DRO, including an implicit bound for general $\phi$ and a more explicit bound for the special case of $\chi^2$-divergence;
>
> 3) Distribution shift setting, including both a general bound, and specialized bounds for MMD DRO and 1-Wasserstein DRO.
>
> 4) We discussed the comparisons of all the bounds above in terms of rate and parameter dependence. Indeed, all these obtained bounds rely only on the true loss function $\ell(f^*,.)$. Moreover, in the case of distribution shift, we especially compared with ERM to argue how such a characteristic of DRO generalization can lessen the "amplification" of the distribution shift experienced by ERM.
>
> We hope that the above addresses the main concern of weakness from the reviewer. The details and proofs of these results can be found in Appendices D, E and F in our revised paper. They are also described in the response to Reviewer J945 above, and we also describe later in this response especially on 1) and 2) that are most relevant to the reviewer's suggestion. Below we first respond to other comments from the reviewer.
>
> ---
> ### Assumption on $\Vert\ell(f^*, .)\Vert_\mathcal{H} \leq M$:
>
> As discussed in Line 256 in our paper, although $\ell(f^*,.)$ is typically unknown and hence we may ultimately resort to using uniformity over $\mathcal L$ to obtain any computable bound, our bound in terms of $\ell(f^*,.)$ explains why DRO can sometimes substantially outperform ERM. As an example, in Section 6, we provide an example  where $\Vert \ell(\theta^*, .)\Vert_\mathcal{H}=0$ and the excess risk of DRO achieves zero with high probability.
>
>
> ---
>
> ### Tightness of Theorem 4.1:
>
>
> Thanks for your good question. Getting a lower bound in general is beyond the scope of this work. However, we have seen at least an instance where the inequality in our bound is an *equality*. This is precisely our example in the numerical section where $\Vert\ell^*(f^*, .)\Vert_\mathcal{H} = 0$. In this case, Theorem 4.1 is easily seen to be tight since our bound gives
> \begin{equation*}
>     P \ell (f^*_{\text{dro}}, .) - P \ell(f^*, .) = 0
> \end{equation*}
> with high probability, which also coincides with our experimental observation.
>
> ---
>
> ### How to resolve MMD DRO computationally?
>
> A very recent AISTATS paper [1] studied tractable reformulations and efficient algorithms to resolve MMD DRO. In particular, their results do not require assumptions on $\ell(f^*, .)\in \mathcal{H}$ and is compatible with the setting of Theorem 5.1.
>
> ---
>
> ### Notation of $P \ell(f, .)$:
>
> $P\ell(f, .)$ is a popular notation in statistical ML and has been adopted by, e.g., the well-known paper [2] on local Rademacher complexities. We mainly use $P\ell(f, .)$ for notation simplicity, but we understand the reviewer's point on mathematical preciseness, and we will use standard notation $E_{ P}[\ell(f, .)]$ in the final version (in fact, we have already used it for our additional materials in the revised paper).
>
> ---
>
> ### Comparison to [3] in distributional shift setting:
>
> We have derived new results for distribution shift in Appendices E and F in our revised paper (as described in the beginning of this rebuttal). Compared with [3] we have the following differences: First, [3] requires structural assumptions of distributional shift, in that $P_\text{test}$ is a projection of $P_\text{train}$ and this projection is known. Our result, on the other hand, only assumes finite distance between $P_\text{train}$ and $P_\text{test}$. Second, [3] studied "absolute bounds" (discussed in our Section 2.1).
> Third, [3] aims to show the statistical efficiency of DRO output, while we establish generalization bounds that depend only on the true loss in distributional shift setting.
> We will clarify our distinctions add comparisons to [3] in the final version.
>
> ---
> ### Explanation on Sobolev Space:
>
> We will clarify the definition of Sobolev space in the final version.
>
> ---
> ### References
> [1] Zhu et al. Kernel distributionally robust optimization: Generalized duality theorem and stochastic approximation. International Conference on Artificial Intelligence and Statistics. PMLR, 2021.
>
> [2] Bartlett et al. Local rademacher complexities. The Annals of Statistics 33.4 (2005): 1497-1537.
>
> [3] Tobias et al. Robust Generalization despite Distribution Shift via Minimum Discriminating Information. Advances in Neural Information Processing Systems 34 (2021): 29754-29767.

---

> > ### Author Response · Authors · 2022-08-02
> > **Moment-Based Ambiguity Sets and Choice of Ambiguity Set**
> >
> > ### Moment-Based Ambiguity Sets
> > The reviewer also asks about moment-based ambiguity sets. While this is a popular class of DRO, they generally lack the *statistical consistency* property that as sample size $n\to\infty$, the ambiguity set can be constructed in such a way that shrinks to a singleton of the true distribution, and thus the obtained DRO solution may not converge to the true optimum. This is because with finite number of moments one cannot identify the distribution (unless the distribution has a small number of known support points). For this reason, we do not pursue moment-based DRO in our extensions.
> >
> >
> > ---
> > ### Choice of Ambiguity Set
> >
> > Regarding the choice of ambiguity set, one way to compare our developed bounds for MMD DRO, Wasserstein DRO, and $\chi^2$-divergence DRO for the purpose of choosing ambiguity set is to look at the dependence on the property of $\ell(f^*,.)$. If $\\|\ell(f^*,.)\\|_{\mathcal H}$ is small, then MMD DRO is preferred. If $\\|\ell(f^*,.)\\|_\text{Lip}$ is small, to the extent that it outweighs the rate deterioration in $n$, then Wasserstein is preferred. If $f$ or $\ell(f,.)$ has a small number of support points, then $\chi^2$-divergence DRO is preferred. Of course, this comparison here may be a bit over-simplistic and needs further experimental testing. Moreover, as the reviewer suggests, subjectivity also could play a role.

---

> > ### Author Response · Authors · 2022-08-02
> > **Extensions to 1-Wasserstein DRO and $\phi$-divergence DRO**
> >
> > ### Generalization Bounds for 1-Wasserstein DRO
> >
> > Suppose that $P$ is a light-tailed distribution in the sense that there exists $a>1$ such that $A:=E_{z\sim P}[\exp \left(\|z\|^{a}\right)]<\infty.$ Then, there exists constants $c_1$ and $c_2$ that only depends on $a, A,$ and $d$ such that for any given $0<\delta <1$, if we choose ball size $\eta = \big(\frac{\log (c_1/\delta)} {c_{2} n}\big)^{1 / \max \{d, 2\}}$ and $n \geq \frac{\log (c_{1}/\delta)}{c_{2}}$, then the 1-Wasserstein DRO solution $f^*_\text{W-dro}$ satisfies
> >
> > \begin{equation*}
> > E_P  \ell (f^*_{W-dro}, .) - E_P \ell(f^*, .)
> > \leq 2\Vert \ell(f^*, .) \Vert_{Lip}
> > \left(\frac{\log (c_1/\delta)} {c_{2} n}\right)^{1 / \max \\{d, 2\\}}
> > \end{equation*}
> > with probability at least $ 1- \delta$.
> >
> > ---
> >
> > For $\phi$-divergence DRO which the reviewer asks about, the analysis is a bit more intricate than Wasserstein DRO due to the absolute continuity requirement in defining the $\phi$-divergence. We first derived a general result:
> >
> >
> >
> > ### General Generalization Bounds for Divergence DRO
> > Suppose that for a given $0<\delta<1$, we construct ambiguity set  $\mathcal{K} = \\{Q: D_\phi(Q \| P_0) \leq \eta\\}$ such that
> > - (high-confidence ambiguity set) $\mathbb{P}[P \in \mathcal{K}] \geq 1- \delta/2 $;
> > - (bound of DRO above the baseline)
> >     $ \max_{Q \in K} E_Q \ell(f^*, \cdot) - E_{P_0} \ell(f^*, \cdot) \leq C_1(\ell(f^*, \cdot), \sqrt{\eta}, \phi)$;
> > - (approximation error)
> >      $E_{P_0} \ell(f^*, \cdot) - E_P \ell(f^*, \cdot) \leq C_2(\ell(f^*, \cdot), n, \delta)$ with probability at least $1-\delta/2$,
> >
> > where $C_1$ and $C_2$ are quantities that depend on the respective arguments. Then, the $\phi$-divergence DRO solution $f^*_{\phi-\text{dro}}$ satisfies
> > \begin{equation*}
> > 	E_P \ell (f^*_{\phi-\text{dro}}, \cdot) - E_P \ell(f^*, \cdot)
> > 	\leq C_1(\ell(f^*, \cdot), \sqrt{\eta}, \phi) + C_2(\ell(f^*, \cdot), n, \delta)
> > \end{equation*}
> > with probability at least $ 1- \delta$.
> >
> > ---
> >
> > This theorem is derived by expanding the analysis for our Theorem 3.1 to incorporate the error between the DRO optimal value over the ball center $P_0$, and also the approximation error induced by $P_0$ over the true distribution $P$. This additional decomposition facilitates the analysis of specific $\phi$-divergences. In the specific case of $\chi^2$-divergence with $P_0$ set to be the empirical distribution, assuming finitely discrete true distribution, we have a more explicit bound:
> >
> > ### Generalization Bounds for  $\chi^2$-divergence DRO on Discrete Distributions
> >
> > Suppose that $P$ is a discrete distribution with $m$ support with $\mathbb{P}[z = z_i] = p_i$. Suppose that (i) $p_{\min} = \min_{1\leq i \leq m} p_i > 0$; (ii) $  \Vert \ell(f^*, \cdot)\Vert_{\infty} < +\infty$.
> > Then, for all $0<\delta<1$, if we choose ball size
> > $$\eta = \frac{1}{n}\Big(m +  2\log(4/\delta) + 2 \sqrt{m\log(4/\delta)}\Big)$$ for  $\chi^2$-divergence DRO, then for all $n \geq \frac{10^6 m^2}{p^3_{\min}\delta^2}$, the  $\chi^2$-divergence DRO solution $f^*_{\chi^2\text{-dro}}$ satisfies
> > \begin{equation*}
> > 	E_{P} \ell (f^*_{\chi^2\text{-dro}}, \cdot) - E_{P} \ell(f^*, \cdot)
> > 	\leq \Vert\ell(f^*, \cdot)\Vert_\infty
> > 	\Big\\{\frac{1}{\sqrt{n}}\sqrt{m + 2\log(4/\delta) + 2 \sqrt{m\log(4/\delta)}} +
> > 	\sqrt{\frac{2\log (2/\delta)}{n}} \Big\\}
> > \end{equation*}
> > with probability at least $ 1- \delta$.
> >
> > ---
> > Note that here we have focused on discrete support because otherwise we would need to use smoothed $P_0$ such as kernel density estimate to account for absolute continuity required by divergence. We believe this latter aspect deserves a substantial future investigation. Nonetheless, both the bounds for Wasserstein and $\chi^2$-divergence DROs rely only on the true loss $\ell(f^*,.)$, in line with the theory for MMD DRO that we have developed.

---

### Official Review · Reviewer_fMCk · 2022-07-10

**Rating:** 5
**Confidence:** 2
**Soundness:** 2 fair
**Presentation:** 3 good
**Contribution:** 2 fair

**Summary:**

In this paper, authors present generalization bounds on the solution from distributionally robust optimization (DRO). In contrast to the hypothesis class complexity in ERM, DRO bounds presented depend on the ambiguity set geometry and its compatibility with the true loss function. They instantiate the DRO bound by using the maximum mean discrepancy DRO distance metric. Their analysis derives generalization bounds that depend solely on the true loss function, independent of any other candidates in the hypothesis class.

**Questions:**

See questions in the detailed weaknesses above.

**Limitations:**

This work has no potential negative societal impact. However, no discussion on limitations or future work is provided. It would insightful if the authors could elaborate on this in the context of the last sentence in the abstract.

**Strengths And Weaknesses:**

**Strengths**:
- Author presents an alternate route avoiding dependency on the complexity class of the function to obtain generalization bounds on the solution from distributionally robust optimization.
- A general DRO bound in Theorem 3.1 is provided before instantiating it in the MMD DRO framework.

**Weakness**:

- The main result in the paper is hard to parse, mainly because of a missing Problem setup section. It would be great if the authors can add a small section describing the notation and the setup very clearly.
- In equation 8, what is the probability measure for the first term in RHS, i.e., what is the measure under which this probability is evaluated $\mathrm{P}(P \not \in \mathcal{K})$?
- While the related work section covers papers on DRO, it misses discussion and references to some other papers where an alternative non-traditional approach has been explored in the context of deep learning[1,2,3]. The generalization bounds presented in these papers also don't directly depend on the complexity of the underlying hypothesis class.
- Extension to MMD DRO is discussed. While the result is interesting and in specific situations tighter than the usual uniform convergence bound. However, the significance and practical relevance of this result are not clear to me. Specifically, it would be insightful to expand on the last sentence in abstract: "...we hope our findings can open the door for a better understanding of DRO, especially its benefits on loss minimization and other machine learning applications."
- Experimental details are missing. While authors redirect the reader to a reference [9] from their paper, authors are encouraged to add more details on the experiments.
- (Minor) Conclusion section with a discussion on limitations and future work is missing.
- Other minor issues or issues that hinder understanding:
  - Theorem 3.1, line 173, What is the first word "DRO"? if it $f_DRO = ...$

[1] Jiang et al. 2021. Assessing Generalization of SGD via Disagreement https://arxiv.org/abs/2106.13799

[2] Garg et al. 2021. RATT: leveraging unlabeled data to guarantee generalization https://arxiv.org/abs/2105.00303

[3] Zhou et al. 2020.  On uniform convergence and low-norm interpolation learning. https://arxiv.org/abs/2006.05942

---

> ### Author Response · Authors · 2022-08-02
> **Response to Reviewer fMCk**
>
> We thank the reviewer for the helpful suggestions. Below is our response:
>
> ---
> ### Problem setup and numerical details:
> We are happy to add more details if the reviewer could refer to which part we are missing, and we would try our best to fill in. Currently, our notations and problem setup are introduced in Section 1, where we define excess risk in Line 23, ERM in Line 26, and DRO in Line 40. For details of numerical experiments, we introduced our hypothesis class in Line 355, how we draw samples in  Line 356, how to choose kernel functions in Line 358, how to reformulate and resolve MMD DRO in Line 360, and our computation environment in Line 363. We believe that our numerical experiment can be reproduced following these details and using our code in supplementary material.
>
> ---
> ### Probability measure for $\mathbb{P}[P\notin \mathcal{K}]$:
>
> According to Eq. (11) at line 196, the ambiguity set $\mathcal{K}$ is centered at $\hat{P}$, which is a random measure. Thus, the probability is over $\hat{P}$ or equivalently the iid data $z_i,i=1,\ldots,n$.
>
> ---
>
> ### Significance and practical usage of our result:
>
> This is a good point and is also asked by the two other reviewers. To address this concern, in the revised paper we have added a substantial expansion of our theory, now in Appendices D, E and F, to cover generalization bounds for the following:
>
> 1) 1-Wasserstein DRO;
>
> 2) $\phi$-divergence DRO, including an implicit bound for general $\phi$ and a more explicit bound for the special case of $\chi^2$-divergence;
>
> 3) Distribution shift setting, including both a general bound, and specialized bounds for MMD DRO and 1-Wasserstein DRO.
>
> 4) We discussed the comparisons of all the bounds above in terms of rate and parameter dependence. Indeed, all these obtained bounds rely only on the true loss function $\ell(f^*,.)$. Moreover, in the case of distribution shift, we especially compared with ERM to argue how such a characteristic of DRO generalization can lessen the "amplification" of the distribution shift experienced by ERM.
>
> We hope that the above alleviates the concerns from the reviewer regarding significance and practicality of our framework. The details and proofs of these results can be found in Appendices D, E and F in our revised paper. They are also described in the response to Reviewer J945 above.
>
> ---
> ### Comparison with non-traditional generalization bounds:
>
> Thanks for bringing up this. At the end of Section 1 (Lines 59 - 71), we compare our results to non-traditional generalization bounds based on hypothesis stability [4], algorithmic robustness [5], and the RKHS norm [6]. Our discussion also applies to [1, 2, 3] that the reviewer referred to. Notice that analysis of these bounds is based on the data-dependent solution $f_\text{data}$ which varies randomly in the hypothesis class. Therefore, to derive clear final bounds, they have to either
> - take a uniform bound over (a subset of) the hypothesis class [3], or
> - take an expectation over $f_\text{data}$, e.g., generalization disagreement [1, Theorem 4.2], hypothesis stability [2, Condition 1].
>
> As a result, these bounds still depend on other candidates in the hypothesis class and thus different from our result. We will clarify this point and add comparisons to [1, 2, 3] in the final version.
>
> ---
>
> ### Conclusion section:
>
> Thanks for your suggestion. We will summarize our limitations and future work in the final version.
>
> ---
> ### References
>
> [1] Jiang et al. Assessing generalization of SGD via disagreement. arXiv preprint arXiv:2106.13799 (2021).
>
> [2] Garg et al. Ratt: Leveraging unlabeled data to guarantee generalization. International Conference on Machine Learning. PMLR, 2021.
>
> [3] Zhou et al. On uniform convergence and low-norm interpolation learning. Advances in Neural Information Processing Systems 33 (2020): 6867-6877.
>
> [4] Bousquet and Elisseeff. Stability and generalization. Journal of Machine Learning Research, 2:499–526, 2002.
>
> [5] Xu and Mannor. Robustness and generalization. Machine Learning, 86(3):391–423, 2012.
>
> [6] Staib and Jegelka. Distributionally robust optimization and generalization in kernel methods. In Advances in Neural Information Processing Systems, pages 9131–9141, 2019.

---

> > ### Comment · Reviewer_fMCk · 2022-08-06
> > **Thank you for your responses.**
> >
> > I thank the authors for their responses. Per the clarifications and responses provided, I have increased my score. While I appreciate the authors' efforts with additional results, because of my limited knowledge in the area, I am unable to appreciate the added significance in practical problems.

---

### Official Review · Reviewer_J945 · 2022-07-11

**Rating:** 6
**Confidence:** 4
**Soundness:** 4 excellent
**Presentation:** 4 excellent
**Contribution:** 3 good

**Summary:**

The paper presents novel bounds that hold for the DRO hypothesis. Interestingly, unlike ERM, uniform bounds are not required and for MMD-DRO the final bound depends only on the optimal hypothesis. Also, unlike local RA bounds, there is no dependence on few other hypothesis.

The bound follows from the key theorem3.1, which though is a simple observation, seems to be special to DRO and does not hold for ERM etc. The final bound for MMD-DRO is presented in theorem4.1. Again, though a simple derivation, is nevertheless interesting.

Apart from the main results, the paper presents other discussions on related bounds, and section 5, where a technical requirement in theorem 4.1 is dropped.

Finally, simulations supporting the theory are presented.

**Questions:**

1. While the methodology proposed is general, details are given only for the MMD-DRO case. Another related case that has been shown to be promising is the Wasserstein-DRO case. Can this case be analysed in detail? Will it also lead to bounds dependent on f^* only? why/why not?

2. Distributional shift is another related case. Can Appendix D be expanded into a main section providing some insights?

**Limitations:**

Yes, insightful discussions are presented in sec4.2.

**Strengths And Weaknesses:**

The paper is very well-written and is a pleasure to read. The result in theorem4.1 is indeed interesting as it has no dependence on any other hypothesis than the optimal.

The bounds also open up many interesting questions, while they explain why DRO performs well in practice. Though the work tries to answer some of them in section5,appD etc., the work would have been far more stronger with a deeper discussion of them. A couple of these questions I summarize in the subsequent section.

Overall, I feel the main strength of the work is the elegant bounds, while weakness is perhaps presentation of a bit fewer results than a good article would contain.

---

> ### Author Response · Authors · 2022-08-02
> **Extensions to other DRO variants and distributional shift setting**
>
> We thank the reviewer for the careful reading and helpful feedback, and are also glad to hear the positive opinion of the reviewer. The reviewer's main concern and questions, are the generalization of our framework to other types of DRO and distribution shift. To respond to this, we have added a substantial expansion of our theory, now in Appendices D, E and F in our revised paper, to cover:
>
> 1) 1-Wasserstein DRO;
>
> 2) $\phi$-divergence DRO, including an implicit bound for general $\phi$ and a more explicit bound for the special case of $\chi^2$-divergence;
>
> 3) Distribution shift setting, including both a general bound, and specialized bounds for MMD DRO and 1-Wasserstein DRO.
>
> 4) We discussed the comparisons of all the bounds above in terms of rate and parameter dependence. Indeed, all these obtained bounds rely only on the true loss function $\ell(f^*,\cdot)$. Moreover, in the case of distribution shift, we especially compared with ERM to argue how such a characteristic of DRO generalization can lessen the "amplification" of the distribution shift experienced by ERM.
>
>
> We hope that the above addresses all the concerns from the reviewer. Below, we describe the added results whose details and proofs can be found in Appendices D, E and F in our revised paper.

---

> > ### Author Response · Authors · 2022-08-02
> > **Comparison to ERM in Distributional Shift Setting**
> >
> > Finally, we would like to point out that the above bounds under distribution shift reveal the potential amplified strengths in using MMD and Wasserstein DROs over ERM under this setting. Namely, if we follow the standard route of analysis for ERM, we would need to upper bound the term $E_{P_\text{test}} \ell (\hat{f}^*, \cdot) -  E_{\hat{P}} \ell (\hat{f}^*, \cdot)$ for ERM solution $\hat{f}^*$, by using a uniform bound. This gives a bound that involves
> > \begin{equation*}
> > \sup_{f\in \mathcal{F}}\Vert l(f, \cdot) \Vert_\mathcal{H} D_\text{MMD}(P_\text{train}, P_\text{test})
> > \end{equation*}
> > or
> > \begin{equation*}
> > \sup_{f\in \mathcal{F}}\Vert l(f, \cdot) \Vert_\text{Lip} D_\text{W}(P_\text{train}, P_\text{test}).
> > \end{equation*}
> >
> > In contrast, the DRO bounds under distribution shift involve only
> > \begin{equation*}
> >     2\Vert \ell(f^*_\text{test},\cdot)\Vert_\mathcal{H} D_\text{MMD}(P_\text{train}, P_\text{test}),
> > \end{equation*}
> > and
> > \begin{equation*}
> >     2\Vert \ell(f^*_\text{test},\cdot)\Vert_\text{Lip} D_\text{W}(P_\text{train}, P_\text{test})
> > \end{equation*}
> > In other words, if the distribution shift is big, to the extent that it outweighs the error coming from the training set noise, then ERM amplifies the impact of distribution shift by a factor that involves a uniform bound on $\{\ell(f,\cdot):f\in\mathcal F\}$, while the corresponding factor in DROs only involves the true loss function $\ell(f^*_\text{test},\cdot)$. This reveals a potentially significant benefit of DRO over ERM in the presence of distribution shift.

---

> > > ### Comment · Reviewer_J945 · 2022-08-08
> > > **thanks**
> > >
> > > I was not able to check the correctness of the new derivations. But they seem to comprehensively address my concerns and improve the contribution. I would like to continue with recommending this paper positively. thanks

---

> > ### Author Response · Authors · 2022-08-02
> > **Extensions to Distributional Shift Setting**
> >
> > Per the reviewer's suggestion, we also investigated the extension of our theory under distribution shift. Here, the data-driven solution $f_\text{data}^*$ is obtained from the training set drawn from $P_\text{train}$, which could be different from the test distribution $P_\text{test}$ where the excess risk is evaluated on.
> >
> > ---
> > ### Generalization Bounds for MMD DRO under Distributional Shift
> > Let $\mathcal{Z}$ be a compact subspace of $ \mathbb{R}^d $, $k: \mathcal{Z}\times \mathcal{Z} \rightarrow \mathbb{R}$ be a bounded continuous positive definite kernel, and $( \mathcal{H}, \langle \cdot,  \cdot \rangle_{ \mathcal{H}})$ be the corresponding RKHS. Suppose also that (i) $\sup_{z \in \mathcal{Z}}\sqrt{ k(z, z) } \leq K$; (ii)  $\ell (f^*_{test}, .)\in \mathcal{H}$  with $ \Vert \ell (f^*_{test}, .) \Vert_{ \mathcal{H}} < + \infty$; (iii) $D_\text{MMD}(P_\text{train}, P_\text{test}) < +\infty$. Then, for all $\delta \geq 0$, if we choose ball size
> > $\eta = D_\text{MMD}(P_\text{train}, P_\text{test}) + \frac{K }{\sqrt{n} } (1 + \sqrt{2\log(1/\delta)})$ for MMD DRO, then MMD DRO solution $f^*_{\text{dro}}$ satisfies
> > \begin{equation*}
> > 	E_{P_\text{test}} \ell (f^*_{\text{dro}}, .) - E_{P_\text{test}} \ell(f^*_{test}, .)
> > 	\leq 2\Vert \ell (f^*_{test}, .) \Vert_{ \mathcal{H}} \left\\{D_\text{MMD}(P_\text{train}, P_\text{test}) + \frac{K }{\sqrt{n} } (1 + \sqrt{2 \log(1/\delta)})\right\\}
> > \end{equation*}
> > with probability at least $ 1- \delta$.
> > ---
> > ### Generalization Bounds for 1-Wasserstein DRO under Distributional Shift
> >
> > Suppose that $d_\text{W}(P_\text{train}, P_\text{test}) < +\infty$ and $P_\text{train}$ is a light-tailed distribution in the sense that  there exists $a>1$ such that $A:=E_{z\sim P}[\exp \left(\Vert z\Vert^{a}\right)]<\infty.$ Then, there exists constants $c_1$ and $c_2$ that only depends on $a, A,$ and $d$ such that for all $\delta \geq 0$, if we choose ball size $\eta = d_\text{W}(P_\text{train}, P_\text{test}) + \big(\frac{\log (c_1/\delta)} {c_{2} n}\big)^{1 / \max \\{d, 2\\}}$ and $n \geq \frac{\log (c_{1}/\delta)}{c_{2}}$, then 1-Wasserstein DRO solution ${f^*}_{W-dro}$ satisfies
> >
> >
> > $E_{P_{test}} \ell ({f^*}_{W-dro}, .)$
> >
> > $- E_{P_{test}} \ell({f^*}_{test}, .) \leq$
> >
> > $ 2 \Vert \ell ( f^*_{test}, .) \Vert_{Lip}  \left\\{ D_\text{W}(P_\text{train}, P_\text{test}) + \Big(\frac{\log (c_1/\delta)} {c_{2} n}\Big)^{1 / \max \\{d, 2\\}}\right\\}$
> > with probability at least $ 1- \delta$.
> >
> > ---
> > Compared to the non-shifted case, the above two bounds involve the amount of distribution shift $d_\text{W}(P_\text{train}, P_\text{test}) < +\infty$ and correspondingly a ball size that is chosen to cover this amount.

---

> > ### Author Response · Authors · 2022-08-02
> > **Extensions to $\phi$-divergence DRO**
> >
> > We also provide generalization for $\phi$-divergence DRO, which is another popular DRO class (this is asked by another reviewer and we believe would add further to the scope of our theory). This case, however, is a bit more intricate than Wasserstein DRO due to the absolute continuity requirement in defining the $\phi$-divergence. We first derived a general result:
> >
> > ---
> >
> > ### General Generalization Bounds for Divergence DRO
> > Suppose that for a given $0<\delta<1$, we construct ambiguity set  $\mathcal{K} = \\{Q: D_\phi(Q \| P_0) \leq \eta\\}$ such that
> > - (high-confidence ambiguity set) $\mathbb{P}[P \in \mathcal{K}] \geq 1- \delta/2 $;
> > - (bound of DRO above the baseline)
> >     $ \max_{Q \in K} E_Q \ell(f^*, \cdot) - E_{P_0} \ell(f^*, \cdot) \leq C_1(\ell(f^*, \cdot), \sqrt{\eta}, \phi)$;
> > - (approximation error)
> >      $E_{P_0} \ell(f^*, \cdot) - E_P \ell(f^*, \cdot) \leq C_2(\ell(f^*, \cdot), n, \delta)$ with probability at least $1-\delta/2$,
> >
> > where $C_1$ and $C_2$ are quantities that depend on the respective arguments. Then, the $\phi$-divergence DRO solution $f^*_{\phi-\text{dro}}$ satisfies
> > \begin{equation*}
> > 	E_P \ell (f^*_{\phi-\text{dro}}, \cdot) - E_P \ell(f^*, \cdot)
> > 	\leq C_1(\ell(f^*, \cdot), \sqrt{\eta}, \phi) + C_2(\ell(f^*, \cdot), n, \delta)
> > \end{equation*}
> > with probability at least $ 1- \delta$.
> >
> > ---
> >
> > This theorem is derived by expanding the analysis for our Theorem 3.1 to incorporate the error between the DRO optimal value over the ball center $P_0$, and also the approximation error induced by $P_0$ over the true distribution $P$. This additional decomposition facilitates the analysis of specific $\phi$-divergences. In the specific case of $\chi^2$-divergence with $P_0$ set to be the empirical distribution, assuming finitely discrete true distribution, we have a more explicit bound:
> >
> > ### Generalization Bounds for  $\chi^2$-divergence DRO on Discrete Distributions
> >
> > Suppose that $P$ is a discrete distribution with $m$ support with $\mathbb{P}[z = z_i] = p_i$. Suppose that (i) $p_{\min} = \min_{1\leq i \leq m} p_i > 0$; (ii) $  \Vert \ell(f^*, \cdot)\Vert_{\infty} < +\infty$.
> > Then, for all $0<\delta<1$, if we choose ball size
> > $$\eta = \frac{1}{n}\Big(m +  2\log(4/\delta) + 2 \sqrt{m\log(4/\delta)}\Big)$$ for  $\chi^2$-divergence DRO, then for all $n \geq \frac{10^6 m^2}{p^3_{\min}\delta^2}$, the  $\chi^2$-divergence DRO solution $f^*_{\chi^2\text{-dro}}$ satisfies
> > \begin{equation*}
> > 	E_{P} \ell (f^*_{\chi^2\text{-dro}}, \cdot) - E_{P} \ell(f^*, \cdot)
> > 	\leq \Vert\ell(f^*, \cdot)\Vert_\infty
> > 	\Big\\{\frac{1}{\sqrt{n}}\sqrt{m + 2\log(4/\delta) + 2 \sqrt{m\log(4/\delta)}} +
> > 	\sqrt{\frac{2\log (2/\delta)}{n}} \Big\\}
> > \end{equation*}
> > with probability at least $ 1- \delta$.
> >
> > ---
> > Once again, the bound involves only $\ell(f^*,\cdot)$ but not other candidates in the hypothesis class. We have focused on discrete support because otherwise we would need to use smoothed $P_0$ such as kernel density estimate to account for absolute continuity required by divergence. We believe this latter aspect deserves a substantial future investigation.

---

> > ### Author Response · Authors · 2022-08-02
> > **Extensions to 1-Wasserstein DRO**
> >
> > Suppose that $P$ is a light-tailed distribution in the sense that there exists $a>1$ such that $A:=E_{z\sim P}[\exp \left(\|z\|^{a}\right)]<\infty.$ Then, there exists constants $c_1$ and $c_2$ that only depends on $a, A,$ and $d$ such that for any given $0<\delta <1$, if we choose ball size $\eta = \big(\frac{\log (c_1/\delta)} {c_{2} n}\big)^{1 / \max \\{d, 2\\}}$ and $n \geq \frac{\log (c_{1}/\delta)}{c_{2}}$, then the 1-Wasserstein DRO solution $f^*_\text{W-dro}$ satisfies
> >
> > \begin{equation*}
> > E_P  \ell (f^*_{W-dro}, .) - E_P \ell(f^*, .)
> > \leq 2\Vert \ell(f^*, .) \Vert_{Lip}
> > \left(\frac{\log (c_1/\delta)} {c_{2} n}\right)^{1 / \max \\{d, 2\\}}
> > \end{equation*}
> > with probability at least $ 1- \delta$.
> >
> > ---
> >
> > As the reviewer rightly suggests, the generalization bound in this theorem depends only on $\ell(f^*, \cdot)$ but not other candidates in the hypothesis class. This theorem is obtained by following our main framework in Theorem 3.1, now using the concentration of $D_\text{W}(P, \hat{P})$ in [1, Theorem 2] and a dual representation for Wasserstein established in [2].
> >
> > Note also that here the rate in terms of $n$ is $n^{-1/\max\\{d,2\\}}$, which is slower than that for MMD DRO. This comes from the large ball size needed to confidently cover the true $P$ and shows a tradeoff between the hypothesis class dependence and the rate in $n$ that differs from ERM. That is, Wasserstein DRO has a generalization that favorably relies only on the true loss $\ell(f^*,\cdot)$, but scales inferiorly in terms of $n$. It is thus most suited when the complexity of hypothesis class outweighs the convergence rate consideration (and is potentially very useful under distribution shift; please see the subsequent response on this aspect).
> >
> > ---
> >
> > [1] Nicolas Fournier and Arnaud Guillin. On the rate of convergence in Wasserstein distance of the empirical measure. Probability Theory and Related Fields, 162(3):707–738, 2015.
> >
> > [2] Peyman Mohajerin Esfahani and Daniel Kuhn. Data-driven distributionally robust optimization using the Wasserstein metric: Performance guarantees and tractable reformulations. Mathematical Programming, 171(1-2):115–166, 2018.

---

> > > ### Comment · Reviewer_J945 · 2022-08-08
> > > **nice**
> > >
> > > Very nice. this will surely make the contribution stronger.

---

### Author Response · Authors · 2022-08-09
**We sincerely thank all the reviewers again for their valuable feedback.**

We sincerely thank all the reviewers again for their valuable feedback. It is encouraging to hear the positive opinions, especially from Reviewers J945 and xNyC who believe our revisions and explanations resolve their main concerns. We also thank Reviewers fMCk and xNyC for raising our scores. We will continue to follow the reviewers' suggestions and update in the final version.

---

### Meta-Review · Area_Chair_pbNS · 2022-09-07

**Recommendation:** Accept
**Confidence:** Certain

**Metareview:**

The paper gives generalization bounds out of distributionally robust optimization (DRO). Unlike standard bounds, e.g., ERM, that depend on the complexity of the function space, the presented DRO bounds depend on the ambiguity set geometry and its compatibility with the true loss function. The authors instantiate their DRO bounds through the maximum mean discrepancy DRO distance, the interesting aspect being the sole dependence on the loss of the best in class hypothesis.
An interesting viewpoint that may spur further investigation in the theory of generalization bounds.




**Award:**

No

---

### Decision · Program_Chairs · 2022-09-14

Accept